# Flare-Ups in Crohn’s Disease: Influence of Stress and the External Locus of Control

**DOI:** 10.3390/ijerph192013131

**Published:** 2022-10-12

**Authors:** María José de Dios-Duarte, Andrés Arias, Carlos Durantez-Fernández, Virtudes Niño Martín, Elena Olea, María Ángeles Barba-Pérez, Lucía Pérez-Pérez, Rosa M. Cárdaba-García, Ana Barrón

**Affiliations:** 1Nursing Department, Faculty of Nursing, University of Valladolid, 47005 Valladolid, Spain; 2Nursing Care Research (GICE), University of Valladolid, 47005 Valladolid, Spain; 3Social Work Department, Faculty of Social Work, Complutense University of Madrid, 28223 Madrid, Spain; 4Health Service of Castilla y León (SACYL), 47007 Valladolid, Spain; 5Instituto de Biología y Genética Molecular (IBGM), Universidad de Valladolid-CSIC, 47005 Valladolid, Spain; 6Social Psychology Department, Faculty of Psychology, Complutense University of Madrid, 28223 Madrid, Spain

**Keywords:** Crohn’s disease, flare-up, stress, external locus of control

## Abstract

(1) Background: The aim of this study was to explore the role of perceived stress and the health locus of control in Crohn’s disease and their influence upon the development of flare-ups of this disease. (2) Methods: Stress and the external locus of control were evaluated in a sample of 64 Crohn’s patients (flare-up phase versus latency phase). The perceived stress scale (PSS-14) and the multidimensional health locus of control scale were the measurement instruments used. (3) Results: The results indicate that the patients have high stress levels during a flare-up (26.13; 27.44; 28.79; 29.67); high stress levels (28.07; 29.67; 27.44; 28.07) if they have a high external locus of control; and that the external locus of control and stress levels have a significant influence upon the existence of flare-ups in those patients with low external locus of control levels (χ2 = 11.127; df = 1: *p* < 0.001). (4) Conclusions: Actions aimed at reducing stress and external locus of control levels are necessary in Crohn’s disease.

## 1. Introduction

Crohn’s disease is a chronic intestinal disease that usually affects any part of the gastrointestinal tract segmentally and discontinuously, occurring more frequently in the terminal ileum, the jejunum and right colon [1,2]. Since this is a recurring disease with flare-up periods, not only the patient’s health is compromised, but also his or her life in general, since this disease is highly incapacitating [3,4]. Although of unknown aetiology, professionals working with Crohn’s patients do, however, observe that flare-ups are related to stressful situations and to how the patient manages his or her illness [5,6].

Psychological stress is a specific type of relationship between the person and the environment, and one of its main effects is the flooding of adaptive capacities, thereby causing an excessive vital overload that exceeds normal self-regulatory possibilities [7,8].

The stress variable has been widely studied by several different authors regarding its relationship to the development of different pathologies: cardiovascular diseases [9,10]; coronary disorders [11,12]; high blood pressure [13,14]; strokes [15]; and brain damage [16,17,18].

Also, stress clearly has a negative influence upon different pathologies [9,19,20]. As with other diseases, the existing research seems to support the theory that the aetiology, course and response to the treatment of digestive disorders tend to be related, among other factors, to high levels of stress [21,22].

The study carried out by Konturek, Brzozowski and Konturek [23] found that stress has an effect upon the brain–gut axis, leading to the release of neurotransmitters and proinflammatory cytokines, affecting the gastrointestinal physiology and thereby causing alterations to gastrointestinal motility, with negative effects upon the intestinal mucosa and microbiota. This is also confirmed in research carried out in 2015 by De Punder and Pruimboom [24], which showed that stress not only increases the symptoms of disease activity, but also that it, alone, provokes a low-level inflammatory state by increasing the availability of water, sodium and energy-rich substances to satisfy the increased metabolic demand induced by the stressor. A review of several works carried out in 2019 also inform of the same [25].

On the other hand, the locus of control, according to Rotter’s original formulation of social learning, should be considered a personality trait [26]. This theory indicates that, when a subject sees something as not completely related to his or her actions, it is then seen as a matter of chance, luck or destiny, as if it were in the hands of other, more powerful people, or caused by a large number of external forces around that person. When an individual interprets things this way, they are categorized as believing in external control. If, however, the individual considers the event to be caused by his or her own behaviour or any other relatively permanent characteristics, then this person is categorized as believing in internal control [26].

Therefore, there are two types of a locus of control: internal and external. The first is characterized by the subject believing that the things happening to him or her are a consequence of his or her actions, and the second is because the subject believes that things occurring are due to external factors [27]. The locus of control is considered as a generalized expectation of problem solving, assuming that behaviours are perceived as an instrument to reach the goal.

The health locus of control, adapted to the clinical situation of patients, is also split into internal or external. In this case, it deals with the person’s idea of his or her capacity to influence his or her own health. Hence, people with an internal health locus of control use more active coping mechanisms, seek information when they are sick, follow treatments better and adopt preventive measures [28]. In contrast, those with an external health locus of control are less active in contributing to improving their wellness [27,29]. It should be noted that these variables are inverse; when an individual has a high external locus of control score, it is low for an internal locus of control and vice versa.

It has also been shown that the locus of control moderates the relationship between stressors and results. Some studies suggest that an external locus of control may be a vulnerability factor [30].

Thus, Barrón establishes that the association between stressful life events and health disorders is greater in subjects with an external locus of control [31].

Kobasa carried out a study in people who had experienced high levels of stress and showed that those who developed disorders scored high in the external locus of control [32].

Roddenberry and Renk [33] found that people with higher levels of stress also have higher levels of disease, as well as higher levels of an external locus of control.

Smith et al. [34] showed that the higher the external locus of control levels were in US combat veterans, the greater their level of general post-traumatic stress disorder symptoms.

A study carried out on patients with multiple sclerosis verified that these patients, just like others with chronic diseases and high levels of an external locus of control, show more maladaptive behaviour, and this is an indication to take into account for psychotherapeutic treatment [35].

One can conclude that both stress, as well as the health locus of control, play an important role in this disease, and many investigations exist from some years ago that relate psychosocial variables to other pathologies [36]. However, scientific evidence regarding the influence of psychosocial factors on Crohn’s patients is scarce [37,38].

Therefore, we believe that the stress variable should be specifically studied in relation to Crohn’s disease and the effect it has, bearing in mind the locus of control variable.

Based on the hypothesis that stress influences the occurrence of Crohn’s disease flare-ups, and that the health locus of control intervenes in said relationship, the aims of this study were to explore the role of perceived stress and the health locus of control in Crohn’s disease and their influence upon the development of flare-ups of this disease.

## 2. Materials and Methods

### 2.1. Participants

An observational study was carried out, specifically a cross-sectional study, which used validated questionnaires to check the influence of stress upon the clinical status of Crohn’s disease patients (a flare-up phase versus a latency phase) and their health locus of control.

The inclusion criterion for the study was that the subjects must suffer from Crohn’s disease and have been officially diagnosed by a digestive specialist. The exclusion criteria were the presence of other physiological illnesses, such as cardiological disease, ulcer, chronic headaches, respiratory diseases, etc., or psychological afflictions such as obsessive disorders, depression, anxiety and so forth. Patients in a flare-up phase were contacted upon their admission to the Digestive Ward of the Gregorio Marañón University Hospital with a diagnosis of a Crohn’s disease flare-up. These patients required hospitalization since they were experiencing cachexia, fever, vomiting and, in some cases, intestinal obstruction, with signs of peritoneal irritation or an intra-abdominal collection. The disease was located along the gastrointestinal tract, except in the upper digestive tract. In all the cases, the patients followed specific daily pharmacological and dietary treatment to control the disease.

The patients in a latency phase were recruited from the Crohn’s Disease and Ulcerative Colitis Association of Madrid. They were asymptomatic and were not being treated with prednisone. These patients said that they felt well and were not experiencing any abdominal pain. They passed 0–2 formed stools a day, without any rectal bleeding.

In all cases, the research was presented personally, together with an explanation of the criteria necessary to be part of the study. Questionnaires were handed out to those patients who expressed their agreement to participate in the study, once it was clear that they had understood everything correctly.

The subjects’ participation in the study was disinterested and voluntary, and, in all cases, informed consent documents were signed. The sample was collected between July 2019 and February 2020.

The study was approved by the Ethics Commission of the Faculty of Psychology of the Complutense University of Madrid (Ref. 2018/19-022).

### 2.2. Indicators

The perceived stress scale (PSS-14) [39] is the Cohen, Kamarck and Mermelstein scale, adapted by Carrobles [40]. The PSS-14 was designed to measure the degree to which life situations are evaluated as stressful.

The Multidimensional Health Locus of Control Scale (MHLC) [41] is the original Wallston, Wallston and Devellis scale, adapted by Garcia-Alcaraz et al. [42]. The scale consists of 18 items, six for each factor, answered according to a Likert-type scale of 6 points, ranging from 0 (“in complete disagreement”) to 5 (“completely agree”).

As regards factors, the following exist:

Factor I. “Other Powerful Factors” (Other Powerful Health Locus of Control): this factor is made-up of the total of the scores obtained for items 3, 5, 7, 10, 14 and 18.

Factor II. “Internal” (Internal Health Locus of Control): this factor is composed of the total of items 1, 6, 8, 12, 13 and 17.

Factor III. “Chance” (Chance Health Locus of Control): this factor comprises the total of items 2, 4, 9, 11, 15 and 16.

All study participants signed informed consent documents and completed the questionnaires in person.

### 2.3. Statistical Analysis

The socio-demographic description of the variables was carried out via an analysis of the percentages, central tendency measures (average) and dispersion (standard deviation).

To explore the role of perceived stress and the health locus of control for well-being in Crohn’s disease, an analysis of the stress levels of patients undergoing a flare-up and those in a latency phase was carried out, dividing the subjects into high and low locus of control groups. For this, the descriptive statistics of mean and standard deviation were used.

To find out the relationship between stress and the locus of control, a specific analysis was carried out that would allow us to observe the relationship between these variables. For this, Pearson’s linear correlation coefficient was used to analyse the linear relationship. This analysis was chosen since the variables studied are interval variables.

With the Pearson correlation analysis, it is possible to establish the relationship that exists between the different variables and see to what extent one of them increases or decreases in relation to the other. However, this type of analysis does not establish a causal dependence. Based on the fact that the dependency relationship was established between an external locus of control and stress in the case of Crohn’s patients in a latency phase, a further study of this relationship was carried out. A binary logistic regression analysis was used for this. This statistical technique aims to test hypotheses or causal relationships when the dependent variable (result) is a binary (dichotomous) variable; that is, it has only two categories.

The influence of stress in predicting the existence of a flare-up was studied, taking into account the external locus of control in these patients. For this, the subjects were divided into two groups, according to their level of the external locus of control, in such a way that they were classified into groups with a low external locus of control and a high external locus of control. The median was used to carry out this classification since it is an unbiased measure of central tendency.

Finally, the area under the receiver operating characteristic (AUROC) was used as an index for the global accuracy of the test.

The Statistical Package for Social Sciences Version 23 (IBM Corp, Armonk, NY, USA) tool was used to carry out the descriptive statistics and hypothesis contrast of the study, as well as the analyses of the internal consistency, reliability and validity of the instruments used in this work. In all the tests, a confidence level of 95% and a *p*-value below 0.05 were considered significant.

## 3. Results

The sample comprised 64 Crohn’s patients with an average age of 34.60 years, of which 30 were men (48.87%) aged between 17 and 50, representing an average age of 35.87 years, and 34 women (53.13%) aged between 16 and 51, representing an average age of 33.41 years.

### 3.1. Relationship between Stress and the Locus of Control in Crohn’s Disease

The results of the analysis of the level of stress in the patients experiencing a flare-up and those in a latency phase, taking into account their classification as having either a high or low locus of control, were significant regarding those with a low external locus of control level. The results are shown in Table 1.

In addition, this table shows that the subjects experiencing a flare-up are also highly stressed, with similar scores, and those with high external locus of control levels, also have high levels of stress.

As regards the Pearson linear correlation coefficient analysis, the two groups of patients were considered separately (flare-up or latency phase). The results are shown in Table 2.

In the variables’ correlation analysis for the group of Crohn’s patients in a latency phase, there was a significant positive correlation (r = 0.565; *p* < 0.01) between the external locus of control and stress variables.

### 3.2. Relationship between the External Locus of Control and Stress, and Its Influence upon Predicting a Flare-Up

Following the proposed hypothesis and the results obtained, we wanted to know the influence of stress in relation to an external locus of control in Crohn’s disease. The influence of stress on the probability of a flare-up was studied, taking into account the external locus of control in these patients. To do so, the subjects were divided into two groups, according to their level of the external locus of control, in such a way that they were classified into groups with a low external locus of control and a high external locus of control. The median was used to carry out this classification, since it is an unbiased measurement of central tendency.

The *t*-test for equality of the means showed that the subjects belonging to the group with a high external locus of control presented higher levels of stress (28.07) than those who belonged to the group with a low external locus of control (15.23).

The knowledge was taken a step further regarding the influence of stress in relation to an external locus of control in Crohn’s disease. A binary logistic regression model was established, in which the binomial and categorical dependent variable was the existence of a flare-up with two options: YES or NO. The introduction method was used for the estimation of the models. Logistic regression is based on the idea that independent variables try to predict the likelihood of something happening or not. 

As regards the analysis of the logistic regression model for the prediction of the probability of a flare-up in the group with a low external locus of control, the main effects model gave, as a result, (χ2 = 11.127; df = 1: *p* < 0.001), turning out to be significant. In contrast, the main effects model was (χ2 = 0.238; df = 1: *p* < 0.625) in the high external locus of control group, which was not significant. The results are shown in Table 3.

Regarding the predictive efficacy of the Crohn’s disease model with a low external locus of control (significant model), the statistics indicated that the estimated model explained approximately 43% of the variability in the dependent variable (flare-up), (R2 de Nagelkerke = 0.426) (Table 4). Its classification table shows a total of 69% correct classifications regarding the existence of a flare-up (cut-off point: 0.05) (Table 5).

Next, in the significant model, the relationship between the independent stress variable and the dependent variable (existence of a flare-up), was analysed in the group with a low external locus of control. The results are shown in Table 6.

It is shown that stress (odds ratio 1.2), is significant for the prediction of a flare-up in the group with a low external locus of control.

As seen in the table, stress explains the fact that the dependent variable (flare-up = YES) occurs (significance less than 0.05), therefore, the greater the stress, the greater the likelihood of a flare-up in these patients. Stress, therefore, is a good predictive variable for the existence of a flare-up.

As a result, the resulting predictive model of the probability of a flare-up in the group of Crohn’s patients with a low external locus of control (Figure 1), would be expressed using the following formula:$$P\left(flare\;up\right)=\frac{1}{1+e^{\left[3.540-\left(0.186*stress\right)\right]}}$$

In addition, the ROC curve was used to measure the effectiveness of the contrasts utilized in the logistic regression analysis (Figure 1).

Through this type of contrast, the accuracy of the prediction of a flare-up is checked using the stress variable in the group of Crohn’s patients with a low external locus of control (significant model).

The ROC curve shifts up and to the left, in such a way that the area below the curve is larger than the area above it. Thus, it is evident that the sensitivity and specificity of the test used, is high.

## 4. Discussion

This paper has helped us learn about the impact that stress, together with the health locus of control, have upon Crohn’s disease flare-ups. As regards stress, the results of our study show that the subjects experiencing a flare-up are highly stressed. These results are consistent with the previous studies carried out regarding other pathologies, in which they state that stress influences the activation of the illness [43,44,45].

This increase in stress can be related to the increase of symptoms and a worse response to treatment [43,46]. A study carried out on patients with irritable bowel syndrome showed that stress is related to the severity of the illness [47]. Also, research by Hirten et al. [48], in 2021, provided evidence that perceived stress, evaluated longitudinally, is significantly associated with the systemic inflammation and symptoms of ulcerative colitis.

The works carried out by Duffy and his team [44] showed an increased risk of active illness in the subjects exposed to stressful situations. A study carried out in 2018 on patients with irritable bowel syndrome also described greater disease activity in the presence of psychological stress [49].

Sewitch and his work group [45] demonstrated that the best predictors of psychopathological symptoms in inflammatory bowel disease correspond to stressful events. The comorbidity of irritable bowel syndrome and psychological anxiety is very common [50]. Evidence exists showing that stress-induced alterations to neuroendocrine-immune pathways act on both the gut–brain axis and the microbiota–gut–brain axis, causing an exacerbation of symptoms in irritable bowel syndrome [51].

These findings are consistent with the relationship between stress and the exacerbation of inflammatory bowel disease [5,23,24], perceived by patients as the main reason for the worsening of their symptoms [6]. The study carried out by Araki in 2020 [38] demonstrates that a worsened mental state positively correlates to disease activity in patients with inflammatory bowel disease, especially in those who believe that their disease is exacerbated by psychological stress. Therefore, the treatment should centre upon controlling stress and stress-induced responses. Research carried out upon patients with ulcerative colitis saw a clear reduction in the disease activity when reducing stress with the practice of yoga [52].

Garcia-Vega and Fernandez-Rodriguez [53] proposed stress management as a treatment for Crohn’s disease patients and verified that these patients’ symptoms improved with stress management training. This was also seen in later studies carried on inflammatory bowel disease patients [54].

Regarding the health locus of control, we were able to verify in our study that the patients who present high levels of an external locus of control also present high levels of stress. These results are consistent with those found in other investigations [31,32,33,34].

The interpretation of this finding leads us to consider that these patients will adapt less effectively to their disease. They will participate less in beneficial health behaviours and active coping strategies, which, in turn, will lead to higher stress levels. This implies that these patients, based on presenting high levels of an external locus of control, stop carrying out actions beneficial to their disease, such as the adherence to pharmacological treatment, diet, physical activity, etc., and this leads to the activation of their disease. Also, being stressed influences their ability to pay attention to their illness, which raises their stress levels further, and influences activation of the disease. So, this poor management of the disease in itself produces a greater level of stress due to its activation, in addition to the influence that the external locus of control may have on stress directly.

The study recently carried out by Krampe and team [55] demonstrated the effect that the locus of control has upon on stress in COVID-19.

Regarding the influence of an external locus of control and stress on the development of flare-ups in Crohn’s disease, it was possible to develop a significant model to predict the likelihood of a flare-up in the case of patients with low external locus of control levels.

These results establish a causal relationship between stress and an external locus of control, and the existence of a flare-up in these patients. This shows the influence that the external locus of control has on stress and, likewise, the influence of both variables on the development of flare-ups in Crohn’s disease.

So, if we were to classify a patient using the health locus of control scale, where they present a low external locus of control, and we knew their stress level (PSS14), we could predict, using the mathematical formula found, the likelihood of this patient experiencing a flare-up (not forgetting, at this point, that it is an estimate in terms of likelihood, rather than mathematical certainty).

Finally, in addition to what has been explained above, we have also shown in this work that two types of effect exist, due to an external locus of control: the direct effect it has on the disease, participating in the patient’s mismanagement of it; and the indirect effect it has by contributing to increased stress levels. At this point, we emphasize that stress has repercussions not only on the patients, but also on the activation of the disease. These results are consistent with those found by the other researchers [32,33,34].

In view of the existing studies and our results, it is very important to point out that, in Crohn’s patients, care and assistance based on the prevention and management of stress, together with a change from an external to an internal locus of control, would contribute to lengthening the latency periods of the disease. Also, the flare-up phases would decrease, with a subsequent reduction in the hospital admissions required by these patients.

Our results allow us to establish a predictive model of the likelihood of a flare-up, dependent upon the level of perceived stress, in the case of patients with a low level of an external locus of control.

The mathematical formula to predict the existence of a flare-up tells us that the higher the stress level, the greater the likelihood of a flare-up. This finding could be used in the conditions specified in this work such as, for example, during consultations, to estimate the likelihood of a patient developing a flare-up and thus initiate preventive treatment to reduce the likelihood of such a flare-up occurring.

## 5. Conclusions

In summary, this study has shown that Crohn’s disease patients experiencing a flare-up are in a highly stressed state. Also, those patients with high external locus of control levels also have high stress levels. In addition, there is a relationship between an external locus of control and stress in Crohn’s disease, and both variables act upon the existence of a flare-up in patients with a low external locus of control. Finally, an external locus of control has a direct effect upon Crohn’s disease and indirect effects that involve an increase in the level of stress and its activation.

Therefore, we propose that the actions prepared for these patients should also include actions that act directly upon stress management by the patients themselves, during any phase of the disease. Also, the actions aimed at influencing the generalized beliefs of the locus of control would be very useful, fundamentally focused on reducing the level of an external locus of control, as well as directed at changing the generalized beliefs of the locus of control from external to internal. These last two interventions are essential when patients are in the latency phase to help them be more receptive towards changing their focus.

On the other hand, it is extremely important that health professionals bear in mind the inclusion of direct actions that reduce stress levels within a patient’s treatment plan. Such actions would help the patients to be more aware of the importance of good disease management within their treatment plan, as well as leading an organized and controlled lifestyle. As a result, they would be capable of facing a stressful and changing environment and better manage their disease, which, in turn, contributes to a more successful treatment of Crohn’s disease.

## 6. Strengths and Weaknesses

The greatest strength of our study lies in having developed a model that shows the influence of both stress and an external locus of control upon the development of flare-ups in some Crohn’s disease patients.

A further strength is having shown that an external locus of control has a direct effect upon the disease and contributes to poor management of the pathology by the patient. It also has indirect effects that are related to an increase in the patients’ stress levels, which, in turn, increase the likelihood of a flare-up.

As regards the weaknesses or limitations of this study, the greatest is the sample size, as well as the type of study proposed. For this reason, our conclusions should be considered with forethought. In our opinion, we feel that these results should be confirmed in further investigations using larger samples, as well as in prospective studies or with another temporal or external cohort.

## Figures and Tables

**Figure 1 ijerph-19-13131-f001:**
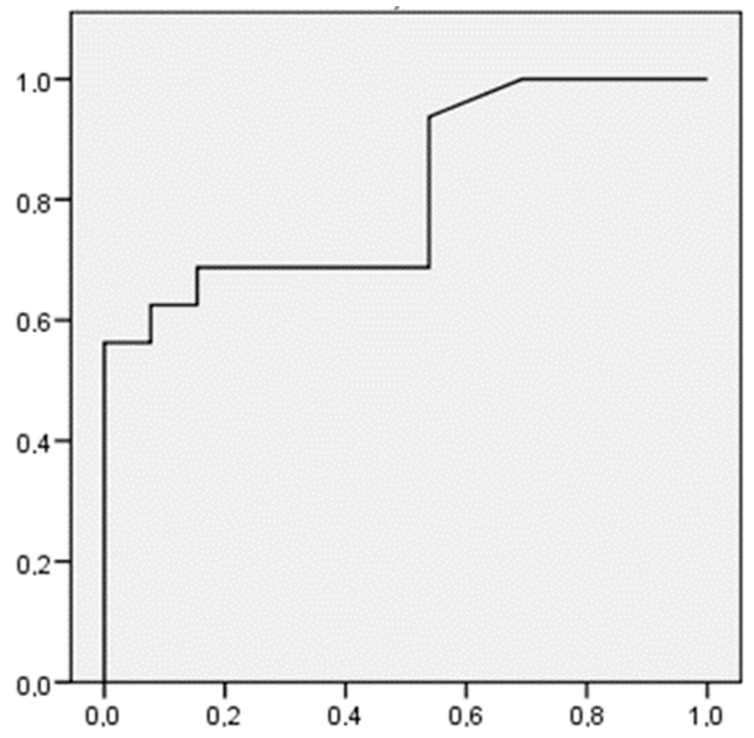
ROC curve for Crohn’s patients belonging to the group with low external locus of control.

**Table 1 ijerph-19-13131-t001:** Stress averages according to the locus of control.

		*N*	Average	SD	*p*-Value
Low external locus of control	Latency	13	15.23	6.858	0.002 *
	Flare-up	16	26.13	9.667	
High external locus of control	Latency	14	28.07	10.126	0.637
	Flare-up	21	29.67	9.43	
Low internal locus of control	Latency	10	19.50	13.302	0.073
	Flare-up	19	28.79	10.201	
High internal locus of control	Latency	17	23.29	9.102	0.186
	Flare-up	18	27.44	9.089	

* *p* < 0.05.

**Table 2 ijerph-19-13131-t002:** Pearson’s correlation matrix considering the groups separately.

		External L. C.	Internal L. C.	Stress
Latency phase	External L. C.	1	−0.109	0.565 *
	Internal L. C.	−0.109	1	0.166
	Stress	0.565 *	0.166	1
Flare-up	External L. C.	1	−0.517 *	0.298
	Internal L. C.	−0.517 *	1	−0.276
	Stress	0.298	−0.276	1

* Correlation is significant at level 0.01 (bilateral).

**Table 3 ijerph-19-13131-t003:** Omnibus Test for the coefficients of the model.

ELC Grp			Chi-Squared	df	Sig.
Low ELC	Step 1	Step	11.127	1	0.001
		Block	11.127	1	0.001
		Model	11.127	1	0.001
High ELC	Step 1	Step	0.238	1	0.625
		Block	0.238	1	0.625
		Model	0.238	1	0.625

**Table 4 ijerph-19-13131-t004:** Summary of the Model.

ELC Grp	Step	−2 Log of Likelihood	R-Squared Cox and Snell	R-Squared Nagelkerke
Low ELC	1	28.765 ^(a)^	0.319	0.426

^(a)^ The estimate ended at iteration number 5 because the parameter estimates changed by less than 0.001 for the ELC Grp = Low ELC segmented file.

**Table 5 ijerph-19-13131-t005:** Prediction of the probability of a flare-up in subjects with a low external locus of control.

External Locus of Control	Observe	Crohn’s Disease Patients		Correct Percentage
		Latency	Flare-up	
Low external locus of control	Latency	9	4	69.2
	Flare-up	5	11	68.8
	Global percentage			69

**Table 6 ijerph-19-13131-t006:** Binary logistic regression model for flare-up prediction.

		B	E.T.	Wald	df	Sig.	Exp (B)
Low external locus of control	Stress	0.186	0.76	6.014	1	0.014	1.205
	Constant	−3.540	1.535	5.317	1	0.21	0.029
	Constant	−0.108	1.104	0.010	1	0.922	0.898

## Data Availability

Not applicable.

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
