# Peer review of "Flare-Ups in Crohn’s Disease: Influence of Stress and the External Locus of Control"

_ijerph, 2022, doi:10.3390/ijerph192013131_

Round 1

Reviewer 1 Report (Previous Reviewer 1)

The authors adequately addressed the reviewers comments and concerns through significant reviews. 

Author Response

Dear Reviewer,

We would like to express our deepest gratitude to you for reviewing our manuscript.

After reading the review report we understand that the process is complete:

Review report (reviewer 1): “The authors adequately addressed the reviewers comments and concerns through significant reviews.”

We greatly appreciate the time taken by the editors and reviewers to provide us with constructive feedback about our manuscript, which was extremely helpful. We would like to extend our thanks to everybody involved.

Yours sincerely,

The authors.

Reviewer 2 Report (Previous Reviewer 2)

The authors have demonstrated that Crohn's disease patients who are going through a flare-up are extremely stressed out. 

Additionally, stress and external locus of control in Crohn's disease are related, and both factors influence the occurrence of flare-ups in patients with a low external locus of control. Last but not least, the external locus of 393 control affects Crohn's disease directly and indirectly by raising stress levels and activating it.

The paper now is better organized and well written. The methodology has been also improved as well as the study results.

If the findings are verified by additional research utilizing larger samples such results might encourage adjustments to the psychotherapy strategy for the particular IBD patient.

Author Response

Dear Reviewer,

We would like to express our deepest gratitude to you for reviewing our manuscript.

After reading the review report we understand that the process is complete:

Review report (reviewer 2): “The authors have demonstrated that Crohn's disease patients who are going through a flare-up are extremely stressed out. Additionally, stress and external locus of control in Crohn's disease are related, and both factors influence the occurrence of flare-ups in patients with a low external locus of control. Last but not least, the external locus of control affects Crohn's disease directly and indirectly by raising stress levels and activating it.The paper now is better organized and well written. The methodology has been also improved as well as the study results.

If the findings are verified by additional research utilizing larger samples such results might encourage adjustments to the psychotherapy strategy for the particular IBD patient.”

We greatly appreciate the time taken by the editors and reviewers to provide us with constructive feedback about our manuscript, which was extremely helpful. We would like to extend our thanks to everybody involved.

Yours sincerely,

The authors.

This manuscript is a resubmission of an earlier submission. The following is a list of the peer review reports and author responses from that submission.

Round 1

Reviewer 1 Report

Thank you for allowing me to review the manuscript by María José de Dios-Duarte and colleagues. In their paper, the authors evaluate the association of stress, locus of control on patients with Crohn’s disease in an attempt to develop a model for predicting disease exacerbations using these predictors.

The paper is interesting and important however I have several concerns.

1)    The number of patients included in this study is very low and the numbers get even lower when subgrouped according to diseae activity level and local of control status. I am not sure the models are adequately powered for appropriate prediction.

2)    I do not feel an accurate model has been developed. I think you have identified an important association between stress and locus of control; however I do not feel you have proven that you can predict disease activity level based on stress and LoC. The title, abstract and body should be adjusted to reflect this re-framing.

3)    The conclusions are way too strong. You have not established a causal relationship (line 136 and 268), rather you have merely identified an association.

4)    In order to validate your claims that your model accurately predicts CD flares, you need to validate it prospectively or in another temporal or external cohort. If you had enough patients, you could split into a train/test cohort.

5)    There are no strengths or limitations sections. There are significant limitations that need to be addressed in order to accurately interpret your findings.

6)    This study is plagued by significant reverse causality. Do we really think stress explains 43% of the variance in the probability of a flare? How do we know that having a flare does not increase stress in those with low locus of control (and those with high external or internal locus of control are better able to cope)? There are so many other complex factors at play.

7)    You need more information about baseline patient characterizes (ex. Location of disease, characteristic, duration of disease, prior treatment, etc)

8)    What steps were taken to ensure patients in the “latency phase” did not have symptoms consistent with active disease (not severe enough to require an admission)?

9)    You should do a better job explaining the clinical implications of your statistical testing. I.e what does an X2= 11.127; g.l.= 1: p<0.001 mean?

10) Abstract results section needs concrete numbers (rather than your interpretation and conclusions) and statistical testing in order to allow readers to draw their own conclusions.

Reviewer 2 Report

Despite these interesting findings, there is a wealth of data that supports the idea that stress plays a role in how Crohn's disease people activate their illness.

Medical professionals must, of course, be aware of the value of direct interventions in lowering stress in people with Crohn's disease and IBD patients in general. However, I don't believe this study adds any new information or broadens public understanding of the topic. Additionally, I advise the authors to stay on subject while commenting on the literature that supports the association between psychological factors and diseases in patients with ovarian cancer, COVID-19, or epilepsy, etc. As a result, unfortunately I believe that this paper doesn't deserve publication in its current form.